# Diastereoisomerically Pure, (*S*)-*O*-1,2-*O*-isopropyli dene-(5-*O*-α-d-glucofuranosyl) *t*-butanesulfinate: Synthesis, Crystal Structure, Absolute Configuration and Reactivity

**DOI:** 10.3390/molecules25153392

**Published:** 2020-07-27

**Authors:** Bogdan Bujnicki, Jarosław Błaszczyk, Marek Chmielewski, Józef Drabowicz

**Affiliations:** 1Centre of Molecular and Macromolecular Studies, Polish Academy of Sciences, Division of Organic Chemistry, Sienkiewicza 112, 90–363 Łódź, Poland; bogbujni@cbmm.lodz.pl; 2Institute of Organic Chemistry, Polish Academy of Sciences, Kasprzaka 44/52, 01–224 Warszawa, Poland; marek.chmielewski@icho.edu.pl; 3Institute of Chemistry, Jan Długosz University, Aleja Armii Krajowej 13/15, 42–200 Częstochowa, Poland

**Keywords:** diastereoisomeric sulfites and sulfinates, sulfinylation, crystal structure, X-ray crystallography, single-crystal diffractometry, absolute configuration, retention and inversion

## Abstract

The reaction of *t*-butylmagnesium chlorides with diastereomerically pure (*R*)-1,2-*O*-isopropylidene-3,5-*O*-sulfinyl-α-d-glucofuranose (*R*)-**4** was found to be stopped at the stage of the corresponding, diastereoisomerically pure 1,2-*O*-isopropylidene-(5-*O*-α-d-glucofuranosyl) *t*-butanesulfinate (*S*)-**10** for which the crystal structure and the (*S*)-absolute configuration was determined by X-ray crystallography. Comparison of the absolute configurations of the starting sulfite (*R*)-**4**, and *t*-butanesulfinate (*S*)-**10** (which crystallizes in the orthorhombic system, space group P2_1_2_1_2_1_, with the single compound molecule present in the asymmetric unit), clearly indicates that the reaction of nucleophilic substitution at the stereogenic sulfur atom in the sulfite (*R*)-**4** occurs with the full inversion of configuration via the trigonal bipyramidal sulfurane intermediate **4c** in which both the entering and leaving groups are located in apical positions.

## 1. Introduction

Racemic and optically active diastereoisomeric and/or enantiomeric sulfinic esters played a very important role in the development of sulfur chemistry and especially sulfur stereochemistry [1,2,3,4,5,6,7,8,9,10,11,12,13,14,15]. Preparation of diastereoisomeric *O*-menthyl *p*-toluenesulfinates **2a/3a** by reacting *p*-toluenesulfinyl chloride **1a** with (-)-(1*R*,3*S*,5*R*)-menthol in the presence of pyridine (Scheme 1) [16], confirmed for the first time stereogenity of a sulfinyl sulfur atom and its optical stability at room temperature [2,3]. It is worth noting that very recently diasteroisomerically pure sulfinate **3a** was isolated in almost quantitative yield by reacting *p*-toluenesulfinyl chloride **1a** with (-)-(1*R*,3*S*,5*R*)-menthol in the absence of a base as an HCl scavenger [17], while all attempts of its isolation under flow conditions resulted in the formation of **2a/3a** mixtures [18].

This sulfinate was used as a precursor of the first enantiomerically pure sulfoxide, i.e., ethyl *p*-tolyl sulfoxide [19,20,21]. Subsequently, the reaction of this diastereoisomeric sulfinates with organometallic reagents has been commonly applied to obtain a very rich family of enantiomerically pure sulfoxides and other sulfinyl derivatives as “Andersen synthesis” [1,2,3,4,5,6,7,8]. Following sulfinylation of the hydroxyl groups of a sugar moiety was used to prepare other diastereomeric sulfinates, which are used again as substrates for the synthesis of enantiomerically pure sulfinyl derivatives. The best results were obtained with diacetone α-d-glucose (DAG) and its cyclohexyl analogue, dicyclohexylidene α-d-glucose (DCG) (Scheme 2) [22,23,24,25,26,27].

In the search for an alternative and new procedure for the preparation of diastereomeric sulfinates derived from readily available sugars, we focused our attention on diastereomerically pure (*R*)-1,2-*O*-isopropylidene-3,5-*O*-sulfinyl-α-d-glucofuranose (*R*)-**4** [28] (Figure 1) and its reactivity with organometallic reagents.

The possibility of using this diastereoisomerically pure sulfite for the preparation of selected diastereoisomeric *O*-sulfinates results from our earlier observation that the reaction of prochiral sulfites with *t*-butylmagnesium chloride could be stopped on the stage of the corresponding sulfinic esters [29] and from the synthesis of enantiomerically enriched *t*-butanesulfinates **6** by the reaction of prochiral sulfites **5** with *t*-butylmagnesium chloride carried out in the presence of optically active aminoalcohols as a chiral complexing agent (Scheme 3) [30].

## 2. Results and Disscusion

Having in the hands diastereomerically pure (*R*)-1,2-*O*-isopropylidene-3,5-*O*-sulfinyl-α-d-glucofuranose (*R*)-**4**, synthesized from α-d-glucofuranose in the sequence of reactions, described earlier by one of us [28], we decided to check if its reactions with Grignard reagents could be stopped at the stage of the corresponding sulfinic acid esters. Reactions of **4** with methylmagnesium iodide **7a**, phenylmagnesium bromide **7b**, and *p*-tolylmagnesium bromide **7c** did not afford optically active sulfinates **8a**–**c**, but symmetrical sulfoxides **9a**–**c** were obtained as products. This can be explained by assuming that precursors of sulfinates **8a**–**c**, magnesium functionalized sulfinates **8′a**–**c** generated in the first stage of the reaction, react immediately with another equivalent of Grignard reagents present in the reaction mixture (Scheme 4). In the case of benzylmagnesium bromide **7d**, 1,2-diphenylethane was isolated as the main product. On the other hand, the reaction of (*R*)-**4** with *t*-butylmagnesium chloride **7e** stopped at the stage of sulfinate **10**. When tetrahydrofuran, diethyl ether and benzene were used as solvents, the product was isolated as a single diastereoisomer, in 88%, 90%, and 85% yield, respectively (Scheme 5).

Analytically pure sulfinate (**-)-**10** was isolated by column chromatography (DCM: Et_2_O 1:1). Its 100% diastereoselective purity was confirmed by conversion (under conditions that preclude racemization of the substrate and product) into the known enantiomerically pure *t*-butyl methyl sulfoxide (+)-(*R*)-**11** (Scheme 6) [25,31]. Given that the configuration at the stereogenic sulfur atom of sulfinate **10** is inverted during this reaction, its (*S*) absolute configuration could be suggested.

It is interesting to note that the reaction of sterically demanded Grignard reagents (such as 1-adamantylmagnesium bromide, 2,4,6-triisopropylphenylmagnesium bromide, or 2,4,6-tri-*tert*-butylphenylmagnesium bromide) with diastereoisomerically pure sulfite (*R*)-**4** did not occur, even if an excess of organometallics was used, and the reaction time at room temperature was prolonged up to 7 days. In these cases, the sulfite (*R*)-**4** was recovered and the Grignard reagent used was converted into the corresponding hydrocarbon during reaction work-up. It should be noted that the recovery of diastereoisomerically pure sulfite (*R*)-**4** indicates simultaneously a lack of racemization under the reaction conditions

To prove without a doubt the absolute configuration of sulfinate (-)-(*S*)-**10** based on the chemical correlation discussed above its structure and absolute configuration at the stereogenic sulfur atom were determined by an X-ray structural analysis (see Figure 2). *(S*)-1,2-*O*-isopropylidene-(5-*O*-α-d-glucofuranosyl) *t*-butanesulfinate (*S*)-**10** crystallizes in an orthorhombic system, in space group P2_1_2_1_2_1_ (Appendix A). The unit cell consists of four monomers. The absolute configuration at the chiral sulfur atom is *S*. The five-membered ring C1,C2,C3,C4,O4 adopts a half-chair conformation, while the ring C1,C2,O2,C7,O1 is an envelope, with the O2 atom being at the flap position. The angle of envelope opening is equal to 29.7°. The calculated values of the asymmetry parameters [32,33] for both rings are collected in Appendix A.

Comparison of the absolute configurations of the starting sulfite, (*R*)-1,2-*O*-isopropylidene-3,5-*O*-sulfinyl-α-d-glucofuranose (*R*)-**4**, and the structure of the corresponding (*S*)-1,2-*O*-isopropylidene-(5-*O*-α-d-glucofuranosyl) *t*-butanesulfinate (*S*)-**10** formed, clearly indicates that nucleophilic substitution at the stereogenic sulfur atom occurs with the full inversion of configuration. This can be explained by the reaction sequence which is shown below in the Scheme 7 and by assumption that the intermediate **4C** has a trigonal bipyramidal structure. According to this mechanistic proposal, discussed in details in our recent paper [34], the observed inversion of configuration is caused by the presence of both the entering *t*-butyl substituent and the leaving *O*-α-d-glucofuranosyl group in the apical positions of this intermediate. This mechanistic proposal (in which we assume the monomeric nature of *t*-butylmagnesium chloride) also explains why the regioisomeric *t*-butanesulfinate **12**, the formation of which, via the intermediate **4E**, is shown in the upper part of Scheme 7 was not observed in all experiments carried out in THF, Et_2_O, nor benzene. This is because the formation of the **4E** intermediate is unfavorable due to the steric repulsion between *t*-butyl group and *O*-α-d-glucofuranosyl group in an equatorial position.

An interesting feature, which is shown in Figure 2, is the presence of a strong O-H...O interaction between oxygen atoms O3 and O11. The distance O3---O11 is 2.888 Å (for other details, see Figure 3). This might be related to the inversion of configuration of the S atom. Besides the intramolecular O3-H(3)---O11 hydrogen bond, there is also an intermolecular O6-H(6)...O3_SYM_ (1+x, y, z) hydrogen bond, building a linear chain parallel to the *a* axis. The distance O6---O3SYM is 2.800 Å (for other details, see Figure 3).

Figure 2 presents the large ellipsoids of the atoms of terminal groups (atom C7 and its attachments C71, C72), most likely due to the increased mobility of that region. This mobility increase seems to be an effect of the involvement of the remaining (stable) region of the molecule in two strong intra- and intermolecular hydrogen bonding systems which result in good stability. Therefore, the regions which are not involved in that H-bonding system are forced to be highly mobile.

## 3. Materials and Methods

All reactions were carried out under argon atmosphere using gun vacuum dried glassware. Dry solvents were degassed before the use. All commercial reagents were purchased from Aldrich (Darmstadt, Germany) and TCI (Tokyo Chemica Industry, Tokyo, Japan). Dry solvents were prepared according to standard procedures. Reactions were followed using thin layer chromatography (TLC) on silica gel-coated plates (Merck 60 F254) with the indicated solvent mixture. Optical rotations were measured on a Perkin-Elmer 241 MC polarimeter (Vienna, Austria) (c = 1). Column chromatography was carried out using Merck 60 (40–63 or 230–400 mesh) silica gel. 1H NMR spectra were recorded at 200 MHz in CDCl3 as solvent. 1H NMR chemical shifts are given in ppm with respect to tetramethylsilane (TMS) as an internal standard.

### 3.1. (R)-1,2-O-Isopropylidene-3,5-O-Sulfinyl-α-d-Glucofuranose (R)-4 [28]

6-*O*-*tert*-Butyldimethylsilyl-1,2-*O*-isopropylidene-α-d-glucofuranose (3.4 g, 10 mmol) in pyridine (10 mL) was treated with freshly distilled thionyl chloride (0.87 mL, 12 mmol). The mixture was kept at 0 °C, and after disappearance of the substrate (24 h) the mixture was poured into water and extracted with DMC. The extract was dried over MgSO_4_, concentrated in vacuo and separated on silica gel column using *n*-hexane-diethyl ether (9:1) as an eluent to give (*R*)-**4** (1.5 g, 45%), m.p. = 54–56 °C, [α]_D_ = +48.0 (c = 1.4 CH_2_Cl_2_), ^1^H NMR (CDCl_3_): δ = 6.01 (d, 1H, *J* = 3.8 Hz, H-1); 4.94 (d, 1H, *J* = 2.6 Hz, H-3); 4.68 (d, 1H, *J* = 3.8 Hz, H-2); 4.46 (t, 1H, *J* = 2.6, 5.5 Hz, H-4); 4.30 (m, 1H, *J* = 4.8, 5.5, 8.0 Hz, H-5); 4.12 (dd, 1H, *J* = 8.0, 10.8 Hz, H-6); 3.93 (dd, 1H, *J* = 4.8, 10,8 Hz, H-6); 1.50, 1.35 (2s, 6H, 2×CH_3_); 0.89 (s, 9H, *t*-Bu); 0.08 (s, 6H, 2×CH_3_).

### 3.2. (S)-1,2-O-Isopropylidene-(5-O-α-d-glucofuranosyl) t-Butanesulfinate (-)-(S)-10

(1 mmol, 0.226 g) of (*R*)-**4** in 10 mL of a solvent (THF, Et_2_O or benzene) was cooled to 0 °C and treated with excess of *t*-butylmagnesium chloride (3 mmol). The progress of the reaction was monitored with TLC. After 4 h for Et_2_O and THF as the solvents (and 6 h for benzene as a solvent), the mixture was worked up with 10 mL of saturated ammonium chloride and extracted 3 times with DCM (3 × 10 mL). Combined extracts were dried over magnesium sulfate and evaporated. Separated on silica gel column using DCM: Et_2_O 1:1 as an eluent gave desired product **10**: 0.291 g (90% for Et_2_O as a solvent), 0.285 g (88% for THF as a solvent), and 0.275 g (85% for benzene as a solvent). M.p. = 60–62 °C, [α]_D_ = −57.8 (c = 1.0 DCM). ^1^H NMR (CDCl_3_): δ = 5.95 (d, 1H, *J* = 3.5 Hz, H-1); 4.56 (d, 1H, *J* = 3.5 Hz, H-3); 4.41 (d, 1H, *J* = 2.2 Hz, H-2); 4.38 (s, 1H, OH); 4.35 (d, 1H, *J* = 3.6 Hz, H-4); 4.21 (dd, 1H, *J* = 2.4, 9.7 Hz, H-5); 4.17 (d.d, 1H, *J* = 2.4, 7.3 Hz, H-6); 4.06 (dd, 1H, *J* = 3.4, 8.7 Hz, H-6); 3.99 (d, 1H, *J* = 3.5 Hz, H-5); 1.55, 1.62 (br, s, 1H, OH); 1.50 (s, 3H, OCH_3_); 1.31 (s, 3H, OCH_3_); 1.23 (s, 9H, *t*-Bu). Anal. Calcd. for C_13_H_24_O_7_S: C, 48.15; H, 7.41; S, 9.87. Found: C, 48.41; H, 7.23; S, 9.99.

### 3.3. t-Butyl Methyl Sulfoxide (+)-(R)-11

A solution of (-)-(*S*)-**10** (0,162 g, 0.5 mmol) in 10 mL of diethyl ether was added dropwise to an etheral solution of methylmagnesium iodide (1 mmol/10 mL) at room temperature. After completion of the addition, the reaction mixture was stirred for 2 h at room temperature. The reaction was then quenched with 10 mL saturated ammonium chloride solution. After being stirred for a while, the layers were separated and the ether phase was extracted twice with water (2 × 10 mL). The combined water phases, after saturation with sodium chloride, were extracted with dichloromethane (4 × 10 mL). The organic solution was dried over magnesium sulfate and evaporated. A crude product was purified on a silica gel column using dichloromethane - ethyl ether (2:1) as eluent affording (-)-(*R*)-*t*-butyl methyl sulfoxide **11** (54 mg, 90%) [α]_D_ = −4.2 (c = 1.0, acetone), 99% e.e.; ^1^H NMR (CDCl_3_): 2.28 (s, 3H); 1.15 (s, 9H). These data are fully consistent with the literature data [25,28].

### 3.4. Reaction of (R)-1,2-O-Isopropylidene-3,5-O-Sulfinyl- α-d-Glucofuranose (+)-(R)-4 with Phenylmagnesium Bromide

A solution of (+)-(*R*)-**4** (0.133 g, 0.5 mmol) in of THF (5 mL) was cooled to 0 °C and treated with a solution of freshly prepared phenylmagnesium bromide (1.5 mmol) in THF (10 mL). The progress of the reaction was monitored with TLC and after disappearance of the substrate (1.5 h) the mixture was treated with aqueous, saturated ammonium chloride solution (7 mL) and extracted with chloroform (4 × 10 mL). The combined organic phase was dried over magnesium sulfate and evaporated. The residue was separated on a silica gel column using DCM: ethyl acetate 2:1 and, finally, with ethyl acetate as eluents for diphenyl sulfoxide **9b** (88 mg, 89%) m.p. 69–71 °C; ^1^H NMR (CDCl_3_): δ = 7.22–7.36 (m, 6H), 7.47–7.62 (m, 4H) (these data are in accordance with literature data [35,36,37]) and 1,2-*O*-isopropylidene-α-d-glucofuranose **13** (71 mg, 65% [α]_D_ = −11.2 (c = 1.2, H_2_O- this value is in accordance with literature data [38]). Substrate (+)-(*R*)-**4** was not detected. A similar reaction of (+)-(*R*)-**4** (0.133 g, 0.5 mmol) with *p*-toluenemagnesium bromide (1.5 mmol) afforded di-*p*-tolyl sulfoxide **9c** (92 mg, 80%) m.p. 92–94 °C; ^1^H NMR (CDCl_3_): δ = 2.35, 7.25, and 7.52 (AB system, *J* = 7.2 Hz, 4H) (these data are in accordance with literature data [39]).

### 3.5. Reaction of (R)-1,2-O-Isopropylidene-3,5-O-Sulfinyl-α-d-Glucofuranose (+)-(R)-4 with 2,4,6-tri-tert-Butylphenylmagnesium Bromide

A solution of (+)-(*R*)-**4** (0.133 g, 0.5 mmol) in THF (5 mL) was treated with a solution of 2,4,6-tri-*tert*-butylphenylmagnesium bromide in THF (1.5 mmol) [40] at room temperature. The progress of the reaction was monitored with TLC. Stirring was continued for 7 days without observing the progress of the reaction, sulfite **4** was still observed. After this time, standard work-up with aqueous, saturated ammonium chloride solution (7 mL) and extraction with DCM (4 × 7 mL) gave an organic solution, which was dried over magnesium sulfate and evaporated. The residue was separated on a silica gel column using ethyl acetate -DCM (2:1) and finally ethyl acetate for the substrate (+)-(*R*)-**4** (104 mg, 78%) and 1,3,5-tri-*tert*-butylbenzene **14** (158 mg, 86%) (^1^H NMR in accordance with literature data [41]). 1,2-*O*-isopropylidene-α-d-glucofuranose **13** (8 mg, 7%) was also isolated.

### 3.6. X-ray Crystallography of (S)-1,2-O-Isopropylidene-(5-O-α-d-glucofuranosyl)t-Butanesulfinate (-)-(S)-10

Crystal and molecular structure of (*S*)-1,2-*O*-isopropylidene-(5-*O*-α-d-glucofuranosyl) *t*-butanesulfinate (-)-(*S*)-**10** was determined using data collected at 100K with an Oxford Instruments KM4 diffractometer using MoKα radiation (Oxford, UK) [42]. The lattice constants were refined by least-squares fit of 17405 reflections collected in the Ø range 3.67–36.14° [43]. A total of 6250 independent reflections were used to solve the structure by direct methods and to refine it by full matrix least-squares using F^2^ [44]. Hydrogen atoms attached to non-carbon atoms were found in a difference Fourier map and refined isotropically. Hydrogen atoms attached to carbon atoms were placed geometrically at idealized positions and refined isotropically, except for the H atoms at the C72 carbon in which Uiso parameters were constrained with their parent atom. Anisotropic thermal parameters were refined for all nonhydrogen atoms. The Coot and Mercury (Cambridge, UK) programs were used for model building and structure visualization [45,46]. The final refinement yielded the values of *R*_1_ = 0.0633 and w*R^2^* = 0.1440 for 263 refined parameters and 5270 observed reflections. The absolute structure was determined by the Flack method [47,48], and the Flack parameter was x = –0.11(9). The calculated parameters for inverted structure, i.e., with assumed opposite (incorrect) chirality, were: *R*_1_(inv) = 0.0643, *wR*^2^(inv) = 0.1561, and x(inv) = 1.06(9). Compound crystallizes in the orthorhombic system, space group P2_1_2_1_2_1_, with the unit cell consisting of four monomers. Crystal data and experimental details are shown in Appendix A. The absolute configurations at the stereogenic atoms are *S*_S1_, *R*_C1_, *R*_C2_, *S*_C3_, *S*_C4_, *R*_C5_.

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
