# Peer review of "Diastereoisomerically Pure, (S)-O-1,2-O-isopropyli dene-(5-O-α-d-glucofuranosyl) t-butanesulfinate: Synthesis, Crystal Structure, Absolute Configuration and Reactivity"

_molecules, 2020, doi:10.3390/molecules25153392_

Round 1

Reviewer 1 Report

See file attached

Reviewer 2 Report

My main concern is about the assignment of absolute structure and absolute configuration. I recommend authors to do it as it is done in alatest iucr journals like Acta Cryst C. 

Apart from the fact that it would be best to measure the structure The Cooper wavelength the number of Bijovet pairs should be given. 

The next point is the quality of the measurements in context of the refinement of the structure. R1 value is relatively high comparing to temperature of measurement and there are large ellipsoids for some atoms. Taking into acount :

_diffrn_reflns_av_R_equivalents 0.0975
_diffrn_reflns_av_sigmaI/netI 0.0604

the integration process should be reperated or more consistent refinemnet of ADP's  should be applied. 

Everything above do not convince me about proper absolute structure.

Round 2

Reviewer 2 Report

Authors improved their manuscript , and convinced me about proper absolute structure. They used other methods and explained it in details.